# Immunogenicity of a spike protein subunit-based COVID-19 vaccine with broad protection against various SARS-CoV-2 variants in animal studies

**Ming-Chen Yang**[1], **Chun-Chung Wang**[2], **Wei-Chien Tang**[3], **Kuan-Ming Chen**[4], **Chu-Ying Chen**[5], **Hsiao-Han Lin**[4], **Yin-Cheng Hsieh**[4], **Nan-Hsuan Wang**[3], **Yin-Chieh Kuo**[5], **Ping-Tzu Chu**[2], **Hsin-Yi Tung**[3], **Yi-Chen Wu**[3], **Juo-Ling Sun**[3], **Sheng-Yu Liu**[5], **Wan-Fen Li**[2], **Wei-Han Lee**[3], **Jiann-Shiun Lai**[5], **Michael Chang**[6†], **Ming-Tain Lai**[7]*

**1** Department of Translational Biology, R&D Division, OBI Pharma. Inc., Taipei, Taiwan, **2** Department of Pharmacology, Pharmacokinetics, and Toxicology, R&D Division, OBI Pharma. Inc, Taipei, Taiwan, **3** Department of Analytics, R&D Division, OBI Pharma. Inc., Taipei, Taiwan, **4** Department of CMC, R&D Division, OBI Pharma. Inc., Taipei, Taiwan, **5** Department of Biologics Discovery, R&D Division, OBI Pharma. Inc., Taipei, Taiwan, **6** CEO Office, OBI Pharma. Inc., Taipei, Taiwan, **7** CSO Office, OBI Pharma. Inc., Taipei, Taiwan

† Deceased.

* mingtainlai@obipharma.com

**Data Availability Statement:** All relevant data are within the paper and its Supporting Information files.

## Abstract

SARS-CoV-2 pandemic has profound impacts on human life and global economy since the outbreak in 2019. With the new variants continue to emerge with greater immune escaping capability, the protectivity of the available vaccines is compromised. Therefore, development a vaccine that is capable of inducing immunity against variants including omicron strains is in urgent need. In this study, we developed a protein-based vaccine BCVax that is consisted of antigen delta strain spike protein and QS21-based adjuvant AB801 in nanoparticle immune stimulation complex format (AB801-ISCOM). Results from animal studies showed that high level of anti-S protein IgG was induced after two doses of BCVax and the IgG was capable of neutralizing multiple variants of pseudovirus including omicron BA.1 or BA.2 strains. In addition, strong Th1 response was stimulated after BCVax immunization. Furthermore, BCvax with AB801-ISCOM as the adjuvant showed significant stronger immunity compared with the vaccine using aluminum hydroxide plus CpG 1018 as the adjuvant. BCVax was also evaluated as a booster after two prior vaccinations, the IgG titers and pseudovirus neutralization activities against BA.2 or BA.4/BA.5 were further enhanced suggesting BCVax is a promising candidate as booster. Taken together, the pre-clinical data warrant BCVax for further development in clinic.

## Introduction

Since the outbreak of SARS-CoV-2 pandemic occurred in Dec. 2019, there are over 757 million confirmed cases and 6.8 million deaths of the world, as of 2023 Feb. The persistence and

**Funding:** The author(s) received no specific funding for this work.

**Competing interests:** The authors have declared that no competing interests exist.

wide spread of this pandemic greatly impact global economy and public health. Tremendous endeavors have been taken to develop vaccines to control the pandemic. Up to date, as many as 34 vaccines have been authorized as either a primary vaccination or booster [1–4]. However, constantly evolution of the virus renders the emergence of several variants in less than two years such as alpha, beta, gamma, delta, and the current dominant omicron strains [5, 6]. Given that most of the COVID-19 vaccines were developed based on wild type strain, the emerged variants are capable of escaping the induced immune responses, thus compromising the protectivity of the vaccines. Lopez Bernal reported that the effectiveness of vaccine after one dose was notably lower against delta variant compared to alpha variant infection [7]. Several studies indicated that, after two doses of vaccination, the serum neutralization titer against delta strain was significantly reduced compared to wild type and significantly lower neutralization activity was detected against Omicron strain [8–10]. These observations clearly demonstrated that the protectivity of current vaccines may not be sufficient for newly emerged variants. A second-generation vaccine targeting variant strain may be an effective way to enhance the protectivity. In this study, we aim to develop a new vaccine that is capable of inducing immunity against variants including delta and omicron strains either as a primary vaccination or a booster.

There are four approaches commonly employed for the development of vaccine by using different type of antigens such as nucleic acid, viral vector, inactivated virus, and protein subunit vaccines. Representative examples for each categories are the mRNA developed by Moderna and BioNTech-Pfizer, adenovirus serotype (Ad26.COV2.S) developed by Janssen, inactivated COVID-19 (BBIBP-CorV) developed by Sinopharm, and spike protein subunit vaccine developed by Novavax (NVX-CoV2373) [11–13]. Among the four types, protein subunit vaccines are known for their advantage on safety and stability. In addition, the immunogenicity of protein-based antigen can be greatly enhanced in combination with a potent adjuvant. Many protein-based vaccines have been developed to fight against SARS-CoV-2 infection. NVX-CoV2373 was developed by assembling trimeric WT strain spike proteins (S protein) to a polysorbate 80 core in combination with nanoparticle adjuvant Matrix-M which is consist of saponin based QS-7 and QS-21 [14]. It showed 92.6% vaccine efficacy against alpha, beta, gamma, epsilon, and iota variants in their phase III clinical trial [15], and has been authorized in multiple countries. Several other protein-based vaccines are also under development, such as CoV2 preS dTM developed by Sanofi-GSK which employs recombinant trimer S protein with trans-membrane region deleted as the antigen with a squalene-based AS03 as the adjuvant, SCB-2019 developed by Clover Biopharmaceuticals which uses S protein trimer as the antigen with AS03 or alum plus CpG as the adjuvant, and ZF2001 developed by Anhui Zhifei Longcom which uses dimeric receptor binding domain of S protein as the antigen with aluminum hydroxide as the adjuvant. These vaccines all demonstrated good protectivity and safety in clinical trials [16–18].

In this study, we aim to develop a protein-based vaccine BCVax that combines delta strain S protein antigen with potent QS-21-based nanoparticle adjuvant AB801-ISCOM (immune stimulation complex). Results from in vivo immunogenicity study in mice showed high level of anti-S protein IgG was induced after immunization. In addition, the induced IgG is capable of neutralizing different variants of pseudovirus infection including delta and omicron strains. Furthermore, strong Th1 response was detected after BCVax immunization based on the ratio of secreted IFN-γ/IL-4 from splenocytes isolated from immunized mice. BCVax with AB801-ISCOM adjuvant induced significantly higher immunity compared to the one with aluminum hydroxide plus CpG 1018 adjuvant or with AB801 adjuvant by showing higher antibody titer induction and T cell responses stimulation. More importantly, when BCVax was employed as a booster, the pseudovirus neutralization activities against BA.2 or BA.4/BA.5

were significantly enhanced. These results suggest BCVax with AB801-ISCOM adjuvant could be a promising new generation COVID-19 vaccine either as a primary vaccination or a booster.

## Materials and methods

### Delta strain S protein expression and purification

The S protein DS2P employed in this manuscript (sequence ID: GenBank QWK65230.1, SARS-CoV-2 lineage B.1.617.2) were mutated at specific sites for abolishment of cellular furin cleavage [19] and for protein stabilization in culture system to ensure steady production. The mutation sites include T19R, G142D, Δ156–157, R158G, L452R, T478K, D614G, P681R, D950N. DS2P was the fusion of ectodomain of delta SARS-CoV-2 spike protein (amino acid residues 14 to 1219) with 2 proline substitutions (K986P and V987P) [20] and 6x His tag. A human rhinovirus 3C protease (HRV3C) recognition sequence was inserted into the sequences between spike protein and His tag to facilitate the removal of His tag after purification. The sequence scheme is shown in Fig 1A. DNA sequence encoding DS2P was codon optimized for CHO cells and synthesized in-vitro by Genewiz. The DNA was then ligated into pcDNA3.4 expression vector (Thermo Fisher Scientific) using NEBuilder DNA Assembly kit (New England Biolabs). The expression vector was then transfected to ExpiCHO-S cells (Thermo Fisher Scientific). The delta S protein trimer was produced by following the instruction of max titer protocol of ExpiFectamine™ CHO Transfection kit (Thermo Fisher Scientific). The secreted DS2P trimer, referred as delta S protein here after, was harvested by collecting supernatant before cell viability dropped below 50%. After centrifugation, the supernatant was stored at 4˚C prior to purification.

Delta S protein was purified by Ni-affinity column using His Trap excel column (Cytiva) and the eluted fractions were pooled and concentrated by a Millipore Amicon Ultra Filter (100 kDa). The purified protein was further treated with HRV3C enzyme (Leadgene Bio) at 25˚C for 16 hours. The His tag fragment was removed, and delta S protein was concentrated by centrifugation using Millipore Amicon Ultra Filter (100 kDa). To assess the protein purity, samples were mixed with SDS loading buffer under reduction conditions and loaded to SDS-PAGE (4 to 12% gradient) followed by staining with InstantBlue.

### Delta S protein purity analysis

The purity of delta S protein was further analyzed using size exclusion chromatography equipped with UV detection (SEC-UV). The SEC-UV protocol was conducted in a HPLC system (Alliance HPLC 2695, Waters) using a SEC column (Sepax, SRT500, 7.8x 300mm) with a single wavelength detection at 280 nm (2996 PDA, Waters). The purified spike protein (10 μg) was analyzed under SEC-UV system with mobile phase of 150 mM PBS solution (pH 7.0) at the flow rate of 0.5 mL/min. The % of purity was determined by comparing the peak area of delta S protein with the areas of rest of other peaks in the chromatogram.

### Binding affinity of delta S protein with human ACE2 receptor

To determine the binding kinetics of delta S protein to human ACE2, biolayer interferometry (BLI) experiments were performed on an Octet instrument (ForteBio, USA). ACE2 protein with Fc tag, purchased from Acro Biosystems, was first captured on Anti-Human IgG Fc Capture (AHC) biosensors and delta S protein was prepared in a serial dilution with assay buffer (0.1% BSA in PBS, 0.05% Tween 20). The final concentrations of S protein were 0.78, 1.56, 3.13, 6.25, 12.5, 25, 50 nM. The association and dissociation constants of ACE2 with the S

A.

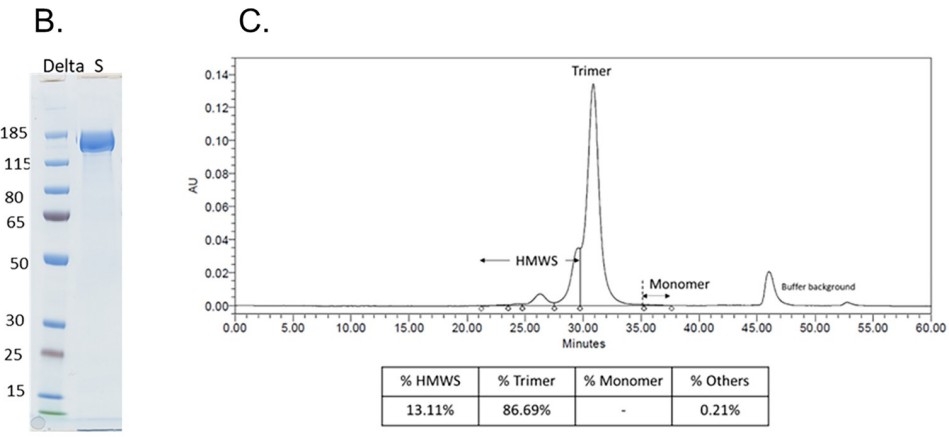

B.                C.

D.

**Fig 1. Characteristics of BCVax delta strain spike protein.** (A) Linear diagram of the delta S protein sequence. The mutation sites of delta strain including T19R, G142D, Δ156–157, R158G, L452R, T478K, D614G, P681R, and D950N were as indicated. (B) SDS-PAGE analysis with Instant Blue staining of delta-S protein under de-naturing conditions. (C) SEC-UV analysis of delta S protein after His-tag removal and purification. (D) Human ACE2 binding activity evaluation of delta S protein using ForteBio Octet analyzer. Data shown in colored lines representing different concentrations of S protein. KD values were calculated using a 1:1 global fit model (Octet). %HMWS: percentage of high molecular weight species.

protein were evaluated for 600 s and 900 s, respectively. KD values were calculated using a 1:1 global fit model using ForteBio's Data Analysis software.

## Preparation of AB801-ISCOM Adjuvant

AB801-ISCOM is consist of QS21-based adjuvant AB801 (Amaran), cholesterol (Sigma), and DOPC (1,2-dioleoyl-sn-glycero-3-phosphocholine) (Avanti). These components were mixed in 5.2:1:2 ratio followed by dialysis to form ISCOM nano-particle matrix as previously described [21]. In brief, phosphatidylcholine and cholesterol were dissolved in chloroform and

then mixed in a glass vial. The resulting solution was transferred to a round bottom flask and the chloroform was evaporated in a rotary evaporator under vacuum followed by addition of AB801 and octyl-beta-glycoside (OG) (Sigma) in tris-buffered saline (TBS). The mixture was then transferred to a shaker and stirred at room temperature until an optically clear micellar solution was obtained. The solution was transferred to a dialysis bag with molecular weight cut-off of 1 KDa followed by dialyzing against 1L TBS to remove OG surfactant. Buffer exchange was performed every 8 to 16 hours for a period of 48 hours [22] to give AB801-IS-COM with an average particle size of 40 nm.

## Electron microscopy (EM) analysis

The delta S protein was prepared in solution of 20mM Tris (pH8.0), 150mM NaCl and AB801-ISCOM was prepared in solution of 145mM TBS (pH 7.4). Delta S protein and AB801-ISCOM were mixed at a ratio of 3:2 followed by dilution to give final the concentrations of delta S protein (20 μg/mL) and AB801-ISCOM (13.3 μg/mL). The EM images were taken by negative stain method. A single drop of sample solution (4 μL) was deposited onto glow-discharged carbon-coated copper grid and washed with deionized water. After staining with 2% (w/v) uranyl acetate for 45 seconds, the solution was removed by filter paper and the grid was placed on bench to allow for air dry before taking images. Images were recorded using a JEM-1400 Transmission Electron Microscope (JEOL) equipped with a field emission electron source under an acceleration voltage of 120 keV at the IMANI center/NCKU (Tainan, Taiwan). Images were taken with a camera system (Model 895; Gatan, Inc. Ultrascan 4000 4K x 4K CCD) at 80,400x magnification and -0.75~1.81 μm nominal defocus.

## Animal immunization

Animal studies were conducted by Level Biotechnology Inc. CRO-Preclinical Testing Center, Taiwan. The procedures with animals described in this protocol have been reviewed by the test facility's Institutional Animal Care and Use Committee. The IACUC approval number was 210306–03 and 220511. During the study period, there is no need to give further anesthesia. At the end of study, all mice were sacrificed by $CO_2$ euthanasia. All procedures described in protocol that involve study animals were conducted in a manner to avoid or minimize discomfort, distress, or pain to the animals, such as providing dietary supplement or 1 mg/kg Meloxicam if necessary.

Briefly, seven weeks old female BALB/c mice (n = 5 per group) were immunized with delta S protein (10 μg) in combination with different adjuvant candidates, including aluminum hydroxide (50 μg) plus CpG 1018 (10 μg), AB801 (5 or 10 μg), and AB801-ISCOM (5 or 10 μg) by intramuscular (IM) injection. For control group, mice were injected with S protein only (10 μg). All mice received two injections on day 0 and day 14. Mice were sacrificed on day 28 post first immunization and the spleen and serum of each mouse were collected for further analysis.

In the immunogenicity study for BCVax as a booster, BALB/c mice were administered delta S protein (10 μg) with adjuvant AB801-ISCOM (2, 5, or 7.5 μg) via intramuscular injection at Day 0 and Day 14. Five out of 8 mice were sacrificed on Day 28 and the rest 3 mice were continue monitored till Day 84 to assess the duration of the immune response. In addition, a group of animals (n = 5) that received two prior injections were given a booster with antigen delta S protein (10 μg) and adjuvant AB801-ISCOM (7.5μg) on Day 56 and sacrificed 4 weeks later on Day 84. Treatment details are shown in Fig 5A. For immunogenicity analysis, blood samples were collected from all animals before each dosing and on Days 14, 28, 56, 70, and 84. Spleens were collected from animals sacrificed on Day 84.

## Mouse sera IgG titer evaluation by ELISA

The anti-S protein ELISA was used to determine the mouse sera IgG titer. All the SARS-CoV-2 S proteins of each variant for coating were purchased from Acro Biosystems. Briefly, SARS-CoV-2 S protein of different variants were coated on the 96-well microtiter plates at 4˚C overnight. After blocked with 1% BSA in PBS, the test samples were diluted with blocking buffer in 2-fold of serial dilution with a dilution factor up to 1:250 followed by adding to the coated microtiter plates and incubated at 25˚C for 60 minutes. After being washed 3 times with PBST buffer, the goat anti-mouse IgG-HRP conjugate was added to the solution. The resulting mixture was incubated at 25˚C for 60 minutes followed by addition of substrate (3,3',5,5'-tetramethylbenzidine) and then stop the reaction with 1N HCl. The absorbance of each well was read at 450 nm in a microtiter plate reader. SARS-CoV-2 variants including wild type, alpha (B.1.1.7), beta (B.1.351), gamma (P.1), delta (B.1.617.2), omicron (BA.1, BA.2, and BA.5) were employed for the evaluation. The titer of each sample was determined by the highest fold of dilution with OD value that was higher than the cut-off point.

## CD4 and CD8 T cell population analysis

The splenocytes harvested from immunized mice were grinded into a single cell suspension. The suspended cells were seeded on 96-well U-bottom plates at density of $2 \times 10^6$ cells/well and stimulated with SARS-CoV-2 S1 mixed peptide pools (2 μg/mL, Mabtech) for 18–20 hours. The resulting cells were harvested for surface and intracellular markers staining. Cells were first incubated with anti-CD4 and anti-CD8 antibodies for surface staining followed by intracellular staining with anti-IFN-γ, anti-IL-4, and anti-Granzyme B antibodies (Biolegend). All flow cytometry data was acquired by Navious EX flow cytometer (Beckman Coulter) and analyzed by Kaluza software.

## ELISPOT cytokine production analysis

Murine IFN-γ, IL-2, and IL-4 ELISPOT were performed following the kit instructions (Mabtech). In brief, splenocytes harvested from immunized mice were grinded into a single cell suspension. The suspended cells were seeded in 96-well ELISPOT plates at density of 250,000 cells/well in duplicates. Cells were stimulated with 0.4 μg/well SARS-CoV-2 S1 mixed peptide pools (Mabtech) at 37˚C for 18–20 hours before counting the spot numbers by Mabtech ASTOR ELISpot reader.

## Neutralization antibody titer evaluation

Pseudovirus production and pseudovirus-based neutralization assay were performed by RNAi Core Facility of Academia Sinica. Briefly, SARS-CoV-2 pseudoviruses expressing full-length spike protein variants including D614G, B.1.1.7, 501Y.V2, P.1, B.1.617.2, BA.1, BA.2 and BA.4/BA.5 were generated by co-transfecting HEK293T cells with pCMV-ΔR8.91, pLAS2w. Fluc.Ppuro, and spike protein encoding plasmid. For neutralization assay, HEK293T-hACE2 cells were seeded with density of $1.0 \times 10^4$ cells/well in 96 well white plates and incubated at 37˚C for 16 to 18 hours. Mouse sera were heat inactivated at 56˚C for 30 min followed by 2-fold serial dilution with DMEM containing 1% FBS and 1x Penicillin/Streptomycin up to a dilution factor of 1:250. The diluted sera were mixed with 1000 transduction unit pseudovirus at equal volumes to give the final sera dilutions at 1:500 followed by incubating at 37˚C for 1 hour. After the incubation, the mixtures were added to the pre-seeded HEK293T-hACE2 cells and incubated at 37˚C for 72 hours. The luminescence emission was triggered by the addition of Bright-Glo-Luciferase reagent (Promega) and the relative luciferase units (RLU) were determined based on the level of luminescence detected by microplate reader (Infinite F500,

Tecan). Pseudovirus only and cells only were employed as the controls for 0% and 100% inhibition, respectively. The 50% inhibition dilution titers (ID50) were expressed as 50% neutralization titer (NT50) and calculated (n = 3) based on the fold of the serum dilution required to obtain a 50% reduction in RLU compared to control. Geometric mean titers (GMT) were determined by GraphPad Prism version 6 software. One way ANOVA with Tukey's multiple comparison test was used to calculate the significance.

## Results

### Characteristics of antigen delta S protein

The design of protein sequence for delta S protein production is shown in Fig 1A. The His-tag sequence was included to facilitate the purification of the protein. In addition, a sequence containing HRV3C cleavage site was inserted into the region between spike protein and His-tag. The cDNA sequence corresponding to delta strain SARS-CoV-2 S protein shown in Fig 1A was cloned into pcDNA3.4 plasmid for transient ExpiCHO cell expression. The secreted S protein was collected through centrifugation followed by purification with Ni-affinity column. The His tag was removed through proteolytic cleavage by HRV3C. The resulting protein was analyzed for its purity by SDS-PAGE. Under denaturing conditions, the S protein showed a major band at 140 KDa, which matches to the molecular weight of monomer S protein (Fig 1B). The protein purity was further assessed by size-exclusion chromatography, the results showed the majority of delta S protein is in trimer form (>85%) with small portion of high molecular weight oligomers (HMWS, <15%) (Fig 1C).

The purified S protein was evaluated for its binding to human ACE2 receptor using Forte-Bio Octet Analyzer. A typical dose-dependent response curves were obtained as shown in Fig 1D. The on-rate and off-rate were determined to be $3.45 \times 10^5$ ($M^{-1}S^{-1}$) and $1.11 \times 10^{-4}$ ($S^{-1}$), respectively, to give a dissociation constant (KD) of 0.32 nM. This result indicates a strong binding affinity of the purified delta S protein to ACE2 (Fig 1D). The binding affinity was 10-fold higher than the findings reported by other study groups with delta strain S protein (KD = 41 nM) [23–25], potentially due to the lower off-rate detected in our delta S protein.

### Characteristics of adjuvant AB801-ISCOM

The ISCOM was prepared by mixing AB801, cholesterol, and DOPC (dipalmitoylphosphatidylcholine) at 5.2:1:2 ratio followed by dialysis with 4 times of buffer exchanges. Homogenous ISCOM nanoparticles were achieved through the process of dialysis. The size of nanoparticle was determined to be 40 nm in average by dynamic light scattering (DLS) with polydispersity index (PDI) of 0.138 suggesting good monodispersity of the nanoparticles (Fig 2A).

Transmission electron microscopy (TEM) was employed to characterize the ISCOM structure. As shown in Fig 2B, a uniform and homogeneous cage-like structure was observed. When delta S protein was mixed with AB801-ISCOM, the majority of S protein was in singular trimer format under TEM as indicated in Fig 2C (upper right), which is consistent with the results from SEC analysis. In addition, the cage-like structure of AB801-ISCOM was maintained in the presence of the S protein as shown in Fig 2C (lower right). The TEM results indicated that there was no interaction between cage-like ISCOM and S protein, and no aggregation was formed in the mixture of S protein and ISCOM.

### Immunogenicity of BCVax in mice

Groups of BALB/c mice were immunized with delta S protein in combination with various adjuvants including aluminum hydroxide plus CpG 1018, AB801, and AB801-ISCOM. The

A.

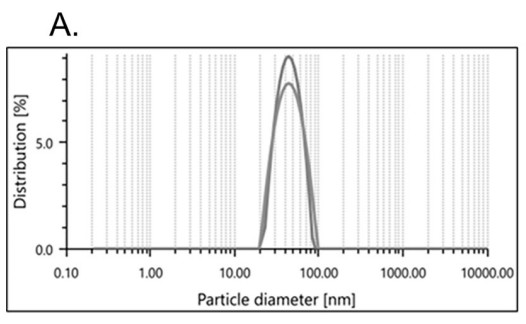

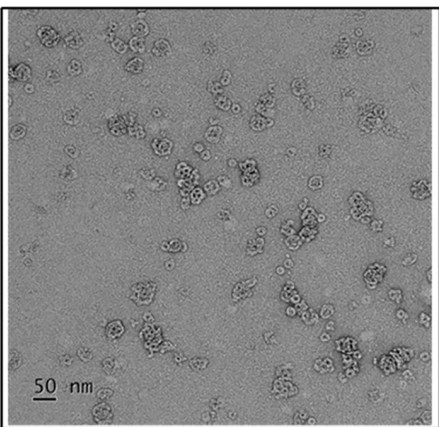

| Parameters | AB801-ISCOM matrix |
|---|---|
| Z-avg diameter (nm) | 43.93 |
| PDI | 0.138 |

B.

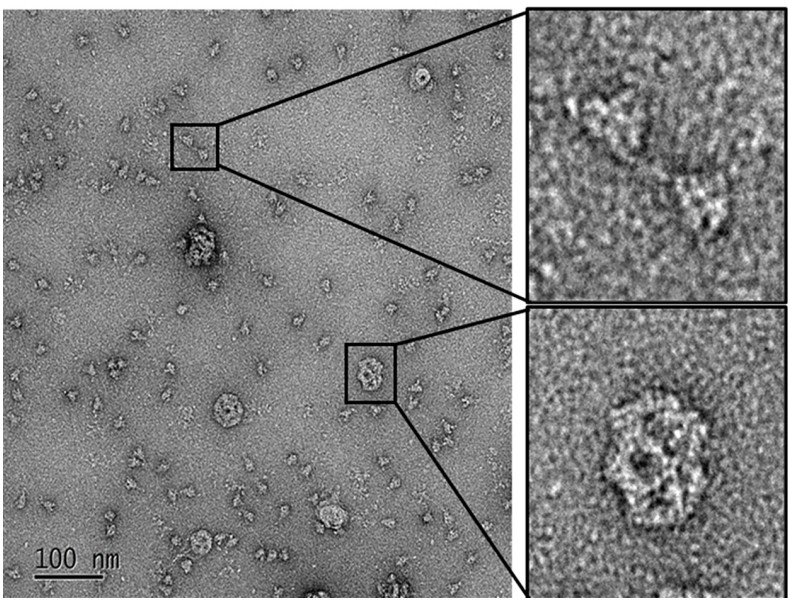

**Fig 2. Characterization of AB801-ISCOM with dynamic light scattering (DLS) and transmission electron microscopy (TEM)s.** (A) Dynamic light scattering (DLS) analysis of AB801-ISCOM. Duplicate analysis was performed to demonstrate the consistency. (B) Transmission electron microscopy (TEM) Image of AB801-ISCOM. (C) The negative stain EM structural analysis of BCVax, consisting of delta S protein and AB801-ISCOM. The EM image showed delta S protein in trimer structure with homogenous distribution. The size of delta S protein and AB801-ISCOM is about 20 nm and 40 nm, respectively. The upper right image shows magnified delta S protein trimer. The lower right image shows magnified AB801-ISCOM in cage like structure.

immunization schedule is shown in Fig 3A. After two injections on day 0 and day 14, serum samples were harvested on day 28 for anti-S protein IgG titer evaluation. The IgG titers against wild type, delta (B.1.617.2), and omicron (BA.1,BA.2, and BA.5) strain S protein were shown in in Fig 3B. The results indicated that AB801-ISCOM adjuvant group shows the highest IgG titers followed by AB801 group and aluminum hydroxide plus CpG 1018 group sequentially. Immunization of delta S protein with adjuvant aluminum hydroxide/CpG 1018 (50 μg/10 μg) elicited IgG titers in $10^5$ range, whereas with AB801-ISCOM (10 μg) as an adjuvant, the IgG titers were in the range from $10^6$ (against omicron strain) to $10^7$ (against delta strain) (Fig 3B).

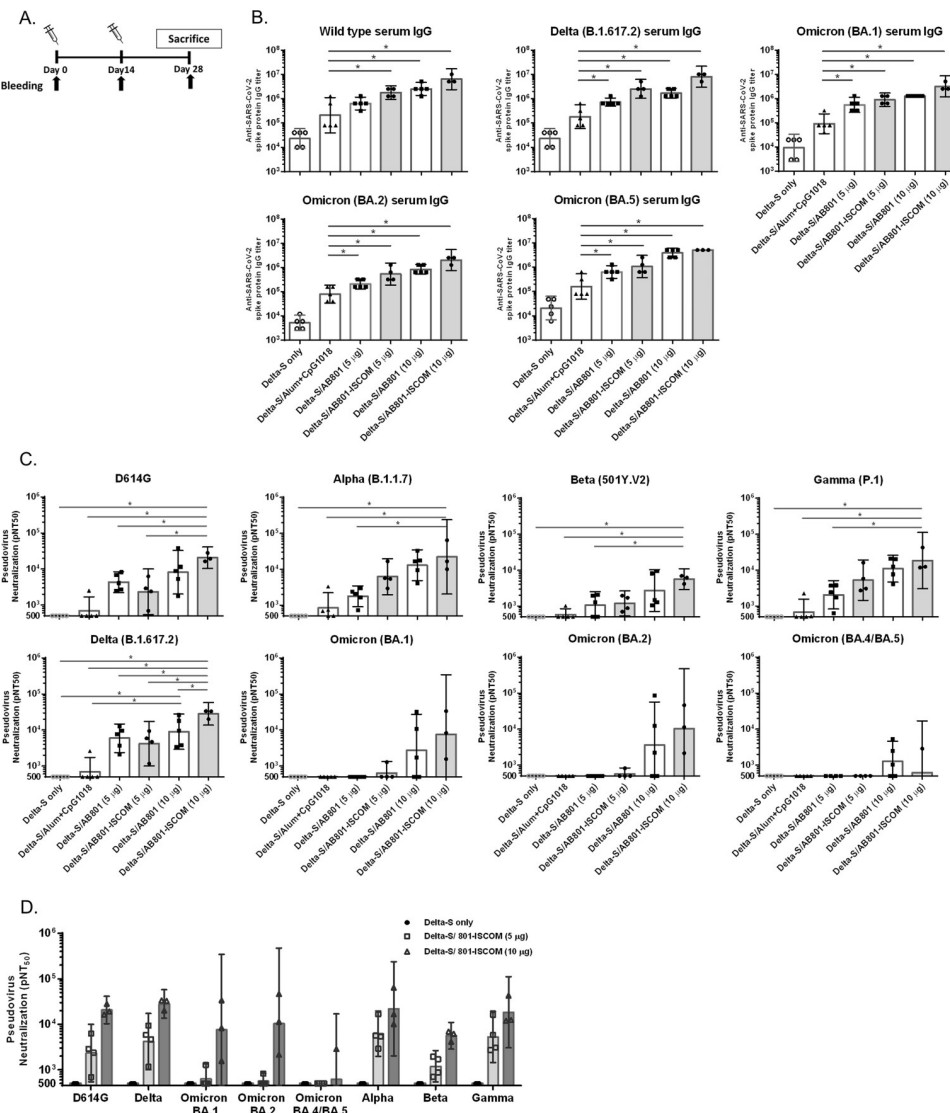

**Fig 3. Evaluation of the immunogenicity of BCVax in BALB/c mice.** (A) Treatment and sampling schedule. The delta S protein (10 μg) combined with or without adjuvant candidates of aluminum hydroxide plus CpG 1018 (50 μg plus 10 μg, triangle), AB801 (5 or 10 μg, square), or AB801-ISCOM (5 or 10 μg, circle, grey bars) were immunized to BALB/c mice on day 0 and day 14. (B) anti-S protein IgG titer of wild type, delta, and omicron strains in immunized mouse serum collected on day 28. (C) Neutralizing antibody titers of sera of immunized mice against SARS-CoV-2 pseudoviruses D614G, B.1.1.7, 507Y.V2, P.1, B.1.617.2, BA.1, BA.2, and BA.4/5 variants. Bars indicate the geographic mean titer (GMT), and the error bars represent 95% confidence intervals. Data were analyzed using one-way ANOVA. (D) pNT50 value of AB801-ISCOM groups against D614G, B.1.1.7, 507Y.V2, P.1, B.1.617.2, BA.1, BA.2, and BA.4/BA.5 variants. ($^*p$ <0.05).

In addition, the effects of adjuvant (AB801 and AB801-ISCOM) on the elicitation of immune responses were shown to be in a dose dependent manner, in which adjuvant at 10 μg elicited higher IgG titers than the adjuvant at 5 μg. When comparing the immunogenicity among treatment groups, the 5 μg AB801-ISCOM adjuvant group showed similar IgG titer to 10 μg AB801 adjuvant group suggesting an approximately two-fold stronger adjuvant activity of AB801-ISCOM compared to AB801 alone, albeit not statistically significant. Consistent findings were also observed in IgG titers against alpha (B.1.1.7), beta (B.1.351), and gamma

(P.1) strains, indicating AB801-ISCOM adjuvanted group showed highest IgG titers compared to AB801 and aluminum hydroxide plus CpG 1018 groups (S1 Fig).

The omicron strain S protein has the highest number of mutations compared to other variants with over 30 mutations giving the omicron viruses with the best capability of escaping current vaccines. As a result, vaccines that were developed with different S protein variants, the effectiveness of the vaccines is compromised against the omicron virus [9, 26]. This phenomenon is also observed in our study, as the anti-omicron S protein IgG titers were about $10^6$ compared to $10^7$ anti-delta or anti-wild type S protein IgG (Fig 3B).

## Pseudovirus neutralization activity

Serum samples harvested on day 28 were also evaluated for neutralization activity against pseudovirus D614G, alpha (B.1.1.7), beta (B.1.351), gamma (P.1), delta (B.1.617.2), and omicron (BA.1, BA.2, and BA.4/5) strains. Neutralization activity was presented as $pNT_{50}$ which represents the potency of the induced IgG at specific fold of serum dilution that can inhibit 50% of pseudovirus infection. The pseudovirus neutralization activity is consistent with the level observed in IgG titers, where the AB801-ISCOM adjuvanted group showed the highest neutralization activity compared to AB801 or aluminum hydroxide plus CpG 1018 across all the variants tested (Fig 3C).

In this study, the neutralization activities of antibodies elicited by BCVax were tested against different variants. Lower neutralization activities were observed with beta and omicron strains, which is similar to previous publications that vaccines are less effective versus beta or omicron strains [9, 27]. However, despite the lowered activity, the neutralization activity against omicron BA.1 and BA.2 variants was still able to achieve nearly $10^4$ range in the AB801-ISCOM adjuvant group at the dose of 10 μg. This result suggests BCVax with delta S protein as the antigen and AB801-ISCOM as the adjuvant may be able to provide protection against most of existing variants including BA.1 and BA.2 (Fig 3D). For BA.4/5 pseudovirus neutralization activity, although the IgG titers for BA.2 and BA.5 are comparable, the activity is much lowered compared to other variants, possibly due to the additional mutation sites of Δ69–70, L452R, and F486V in BA.5 strain. The results showed BCVax induced the strongest neutralization activity against delta strain, followed by alpha, D614, gamma, omicron BA.1, BA.2, beta, and BA.4/5 strain in the order of potency.

## T cell responses

In addition to assessing the antibody titers and neutralization activity, we also evaluated the T cell responses after BCVax immunization. The splenocytes harvested from immunized mice on day 28 were analyzed for CD4 and CD8 T cell population. For CD4 cells, compared to aluminum hydroxide plus CpG 1018 group, the CD4$^+$IFN-γ$^+$ population was significantly increased at dose of 10 μg for both AB801 and AB801-ISCOM adjuvant groups. For CD4$^+$IL-4$^+$ population, no differences were noted across the treatment groups. AB801-ISCOM groups were found to induce CD4$^+$IFN-γ$^+$ and CD4$^+$IL-4$^+$ cell population in a dose-dependent manner, but not for AB801 groups (Fig 4A). For CD8 T cell, compared to aluminum hydroxide plus CpG 1018 group, both AB801 and AB801-ISCOM adjuvant groups (5 and 10 μg) showed significant higher level of CD8$^+$IFN-γ$^+$. In addition, AB801-ISCOM adjuvant group (10 μg) showed greater enhancement of CD8$^+$IFN-γ$^+$ population than AB801 group (10 μg). Furthermore, a marked high level of CD8$^+$GranzymeB$^+$ cells was detected in AB801-ISCOM adjuvant group (Fig 4B), whereas no enhancement of CD8$^+$GranzymeB$^+$ cells was observed with AB801 group. Taken together, these results suggest that ISCOM plays an important role in triggering T cell responses.

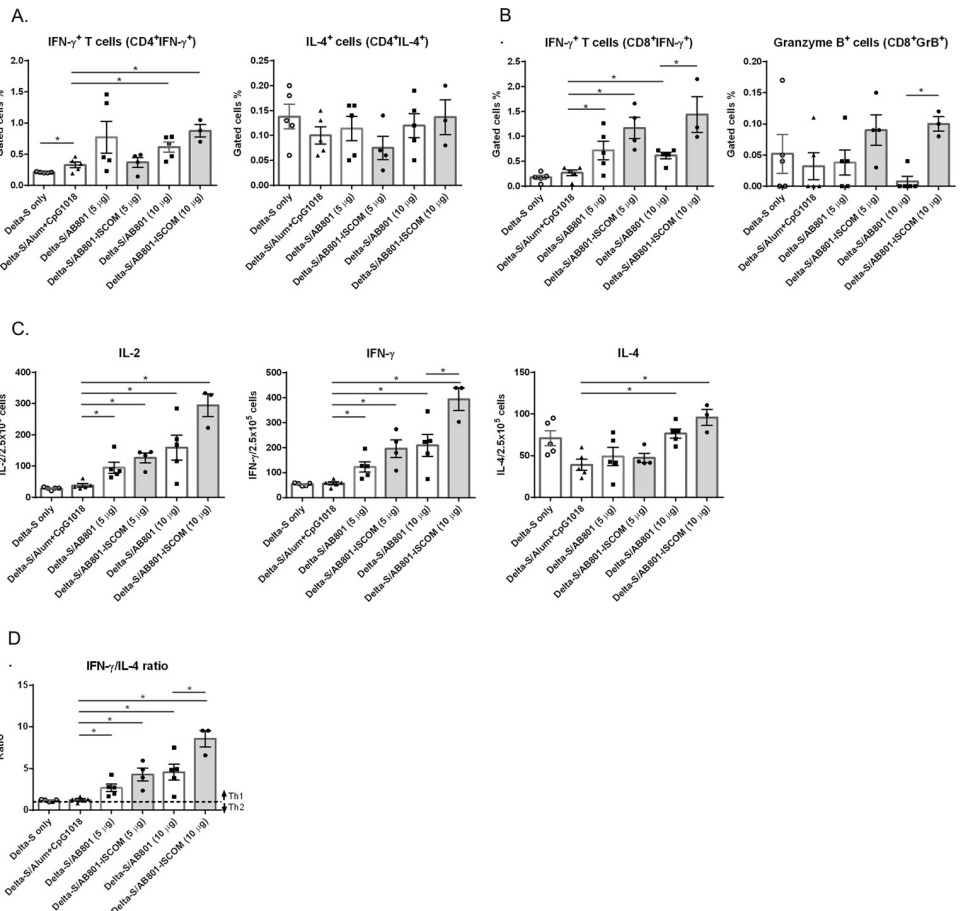

**Fig 4. T cell responses in BCVax immunized BALB/c mice.** The delta S protein (10 μg) combined with or without adjuvant candidates of aluminum hydroxide plus CpG 1018 (50 μg plus 10 μg, triangle), AB801 (5 or 10 μg, square), or AB801-ISCOM (5 or 10 μg, circle, grey bars) were immunized to BALB/c mice on day 0 and day 14, splenocytes were harvested on day 28 for analysis. (A) Flow cytometry analysis of CD4 T cell populations of CD4+IFN-γ+ and CD4+IL-4+ cells. (B) CD8 T cell populations of CD8+IFN-γ+ and CD8+GranzymeB+ cells were evaluated using immunized mice splenocytes. (C) The number of IL-2, IFN-γ, and IL-4 secreting cells from immunized splenocytes per 2.5E+5 cells were analyzed by ELISPOT assay. (D) The ratio of IFN-γ/IL-4 of ELISPOT results. Bars represent mean ± SD. Statistically significant differences were indicated (*$p < 0.05$).

Given the crucial role of Th1 and Th2 in the immune responses, in which Th1-type cytokines tend to produce the proinflammatory responses responsible for killing intracellular pathogens and Th2 cells are implicated in the defense against extracellular pathogens. Th1 effector cells are characterized by the production of inflammatory cytokines such as IFN-γ, TNF-α, and IL-2, capable of stimulating the macrophages, the NK cells, CD8+ cytotoxic T cells. Th2 effector cells are characterized by the production of anti-inflammatory cytokines such as IL-4, IL-5, and IL-13. For an effective vaccine, it is important to have high level Th1 responses to induce cytotoxic T cells.

As a result, we next analyzed cytokine production profiles. Splenocytes harvested on day 28 were stimulated with S protein peptide pool and the level of for IL-2 (Th1), IFN-γ (Th1), and IL-4 (Th2) secreted from cells were evaluated by ELISpot assay. The result showed that AB801 or AB801-ISCOM adjuvant groups induced IL-2 in a dose dependent manner and the level is higher compared to aluminum hydroxide plus CpG 1018 adjuvant group. Among the treatment groups, with the splenocytes from the dose of 10 μg, AB801-ISCOM group showed the

highest level of IL-2 induction with around 2- and 8- folds higher than AB801 group and aluminum hydroxide plus CpG 1018 group, respectively. Similar pattern was observed with the IFN-γ level, where AB801-ISCOM (10 µg) induced the highest level of IFN-γ among the treatment groups (approximately 2 folds higher than AB801 group), suggesting the potent effects of ISCOM on the induction of Th1 responses. In addition, the IL-4 ELISpot results also showed that the spot numbers were higher with the splenocytes from the dose of 10 µg for both AB801 and AB801-ISCOM adjuvant groups compared to aluminum hydroxide plus CpG 1018 adjuvant group, yet the extend of enhancement was not as high as IL-2 and IFN-γ (Fig 4C). To further assess the polarization of T cell response (Th1 or Th2) of the BCVax candidates, the ratio of IFN-γ/IL-4 of ELISpot was determined as shown in Fig 4D. When the ratio is greater 1, the vaccine is considered to induce greater Th1 responses than Th2 responses. In this study, both AB801 and AB801-ISCOM at 10 µg showed the ratio to be 4 and 7, respectively, indicating both types of adjuvants can induce significant higher level of Th1 responses.

## Impact on the immunogenicity of BCVax as a booster

Since majority of global population have taken at least two doses of COVID-19 vaccination, BCVax was also evaluated for the potential as a booster. In addition to the groups of mice receiving two injections on day 0 and day 14, a group of mice with a third injection on day 56 was included in the study. The dosing scheme is shown in Fig 5A. Five out eight mice which received two vaccinations were sacrificed at day 28 and the rest of three mice were maintained for continue monitoring the IgG titer and neutralization activity to assess the durability of immune responses after two doses of vaccination. Serum samples from each group were harvested on the schedule shown in Fig 5A for evaluation of anti-S protein IgG titer and neutralization activity against existing variants. The group with booster showed elevated IgG titers against delta and BA.2 strains, more importantly the IgG titer against BA.5 strain was significantly enhanced on day 84 compared to mice only received two injections (Fig 5B). The time course of the IgG titer was shown in Fig 5C. For the mice receiving two injections, IgG titers reached peak level at week 4 and sustained at the same level until sacrifice at week 12 (Fig 5C). In the mice receiving a booster, the IgG titers were further stimulated to higher levels, not only to delta and BA.2 strains, but also to BA.5 strain (Fig 5B and 5C).

The pseudovirus neutralization activity of booster was also evaluated against delta (B.1.617.2) and omicron strains (BA.2 and BA.4/BA.5) using serum samples collected on Day 84. Consistent with the findings on IgG titers, the booster elicited higher neutralization activity than the group with only two injections as shown in Fig 5D. The neutralization activity was not statistically different against delta strain either with or without booster. However, although the neutralization activity against BA.2 or BA.4/BA.5 strains was limited in mice only received two injections, the neutralization activity reached $10^4$ level with the mice received the booster (Fig 5D). For the T cell population analysis, the CD8$^+$IFN-γ$^+$ and CD8$^+$GranzymeB$^+$ populations were also enhanced after receiving the booster (Fig 5E). Taken together, these results suggest BCVax has a promising potential as a booster with a broad coverage against various variants.

## Discussion

COVID-19 delta strain began to emerge in the late 2020 and quickly became the most deadly variant of the virus. It is reported to be less sensitive to convalescent or vaccination induced antibodies [28–31]. With the emergence of omicron strain in late 2021, it quickly became the dominant strain worldwide. Despite the fast spread, most of the omicron cases were found to have mild symptoms, while the delta strain is still the strain poses the highest risk of disease

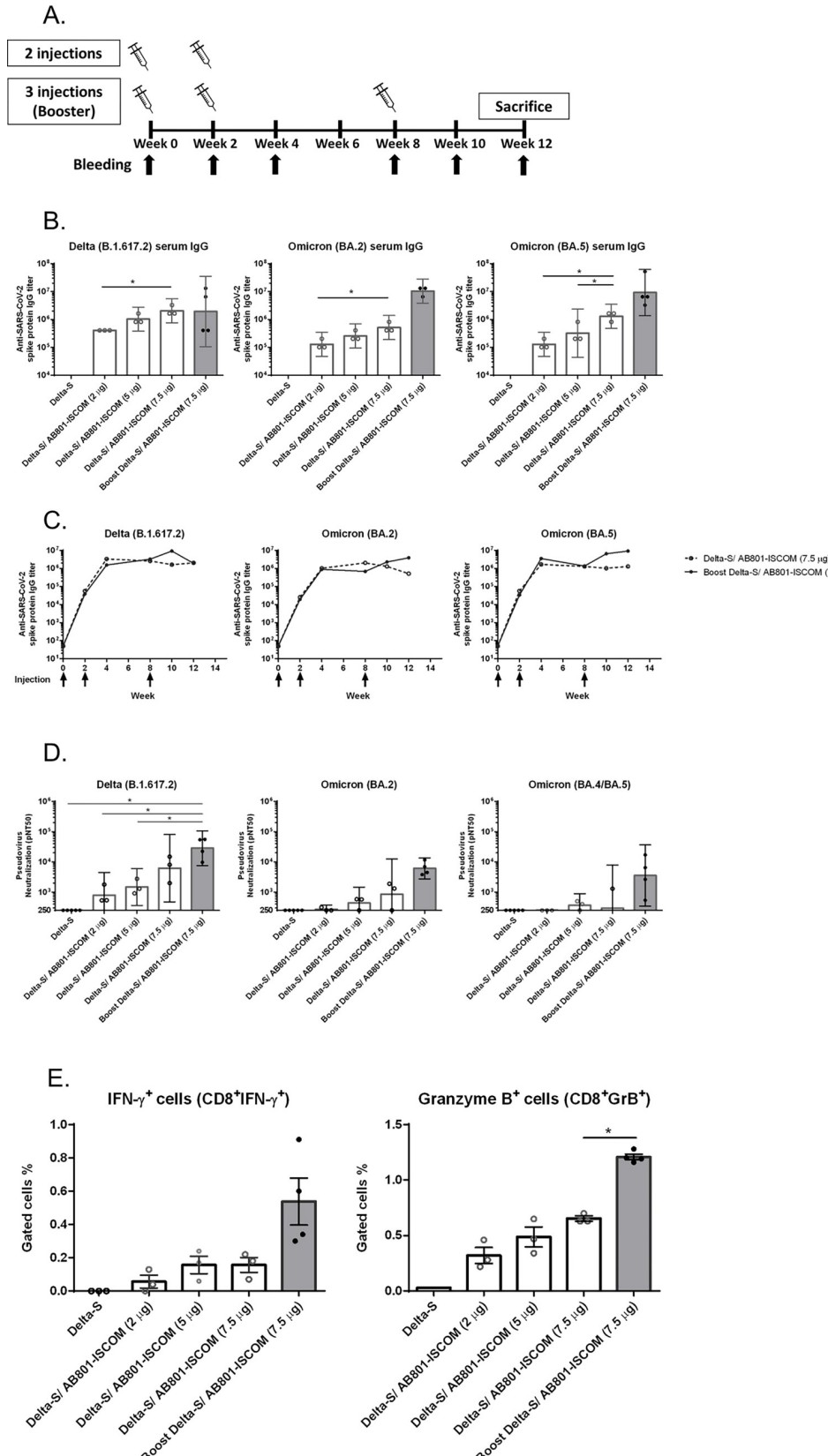

**Fig 5. Evaluation of the immunogenicity of BCVax candidates in BALB/c mice with booster injection.** Groups of mice were immunized with 10 μg of delta S protein combined with different dosage of AB801-ISCOM (7.5, 5, or 2 μg) on day 0 and day 14 (white bars). Another group of mice were received first two injections of 10 μg delta S protein with 7.5 μg AB801-ISCOM on day 0 and day 14, and a booster of same dosage on day 56 (gray bars). (A) The treatment schedule and sampling time. (B) anti-S protein IgG titers of delta, omicron BA.2 and BA.5 strains in immunized mouse serum collected on day 84. (C) The time course of serum IgG response. (D) pseudovirus neutralizing activity of sera collected at day 84 of immunized mice against SARS-CoV-2 pseudoviruses B.1.617.2, BA.2, and BA.4/BA.5 variants. Bars indicate the GMT, and the error bars represent 95% confidence intervals. Data were analyzed using one-way ANOVA. (E) CD8 T cell populations of CD8$^+$IFN-γ$^+$ and CD8$^+$GranzymeB$^+$ cells were evaluated using immunized mice splenocytes harvested on day 84. Bars represent mean ± SD. Statistically significant differences were indicated ($^*p < 0.05$).

severity [32–34]. Another potential threat of delta strain comes from the possibility of "delta-cron" strain to emerge, especially an actual deltacron case was reported in China in Apr 2022 [35]. A study conducted in Israel indicated that the crypt circulation of delta strain during omicron dominant period [36]. In this study, the delta strain S protein is selected as the antigen of BCVax. More importantly, we combined it with a potent nanoparticle adjuvant AB801-ISCOM. The activity of the vaccine in triggering neutralization antibody and T cell responses was evaluated in mouse model. In comparison to commonly used adjuvant aluminum hydroxide plus CpG1018, BCVax showed significantly higher serum IgG titer, better pseudovirus neutralization activity against multiple variants, and stronger Th1 responses after two doses of immunization. When employed as a booster, the neutralization activity of induced antibodies was further elevated, especially against BA.5 compared to mice with only two vaccinations, supporting the broad coverage of BCVax against multiple variants. Although BCVax development is based on delta strain S protein, we found that the in vitro neutralization was not limited to delta strain only, with two or three injections of BCVax, the in vitro neutralization is effective to omicron strains as well. These results suggest that potential application of BCVax as a booster to enhance protectivity against all current existing variants.

The immunogenicity of COVID-19 vaccines was mostly evaluated via antibody response, however, accumulating evidence has shown that T cell responses induced by vaccination is as important as antibody induction [37, 38]. Tan et al. reported that early induction of IFN-γ-secreting SARS-CoV-2-specific T cells can accelerate viral clearance in patients with mild disease suggesting the importance of T cell response during SARS-CoV-2 infection [39]. Study from Bilich et al. revealed that T cell responses in convalescent individuals persist longer than serum IgG indicating the long-term protectivity of T cell immunity [40]. It is reported that T cell memory encompasses broad recognition of viral proteins, the breadth of recognition can limit the impact of individual viral mutations and is likely to protect subjects from severe diseases form viral variants [41]. In our preclinical study of BCVax, the T cell responses in AB801-ISCOM adjuvant groups showed significantly higher level of CD4$^+$IFN-γ$^+$ and CD8$^+$IFN-γ$^+$ population induction compared to aluminum hydroxide plus CpG 1018 or AB801 groups. The ELISpot results also showed Th1-prone immunity is significantly induced with AB801-ISCOM adjuvant (Fig 4). The nanoparticle characteristic of the adjuvant AB801-ISCOM may contribute to the T cell population induction. The average particle size in 40 nm of ISCOM is favorable for antigen presenting cells to uptake, hence stronger immunogenicity can be induced compared to larger size particulate adjuvant [42, 43]. It is reported that ISCOM induces Th1 responses (i.e. IFN-γ and IL-2) more prominent than Th2 responses in the studies with Epstein-Barr virus (EBV) and human immunodeficiency virus (HIV) [44–46] giving better protectivity against the viruses. Consistent with previous reports that ISCOM as an adjuvant could promote antibody responses, as well as T helper and cytotoxic T cell responses in many animal models [43, 47], our study provides further evidence that BCVax is capable of inducing high level of T cell responses and potentially offers protection against COVID-19 variants.

In comparison among different types of vaccines, Zhang et al. has published a study in 2022 to compare the humoral and cellular immune memory among four types of COVID-19 vaccines in human [48]. The results showed that at 6 months post immunization, antibody levels declined, while memory T cells and B cells were comparatively stable, and similar levels of CD4 and CD8 T cell responses were observed between mRNA and protein-based COVID-19 vaccines. Hielscher et al. published their study demonstrating NVX-CoV2373 vaccination significantly induces spike-specific antibodies and CD4 T-cells, albeit at lower levels than BNT162b2 and mRNA-1273 vaccines. More importantly, the cross reactivity of CD4 T-cells towards the parental strain and all VOCs confirms the promising protectivity of the protein-based vaccine [49]. Although the emergence of omicron strain began in late 2021, it quickly became the dominant strain worldwide. Despite most of the omicron cases were found to have mild symptoms, the omicron strains continue to evolve and spread quickly [50]. Therefore, we also evaluate BCVax against omicron strains. For mice received two injections, the vaccination induced sustainable antibody response in which the anti-S protein IgG titers against delta and omicron BA.2 or BA.5 variants maintained at $10^6$ level until the end of study period. The enhancement effect of the booster is more profound in BA.2 and BA.5 IgG titers, where delta IgG titers were in $10^6$ range with or without a booster (Fig 5C). This observation is in line with previous reports that booster can provide enhanced immunity against different variants, based on the rationale of boosters are capable of expanding the preexisting SARS-CoV-2 S-specific memory B cells, as well as de novo induction, leading to enhanced potency and breadth [51, 52].

The bivalent booster vaccine BNT162b2-Omi4/5 developed by Pfizer/BNT was granted authorization recently, based on the preclinical study results. The pseudovirus neutralization activity against BA.1, BA.2, BA.2.12.1, and BA.4/5 reached $10^3$ to $10^4$ level from mice received two BNT162b2 injections followed by BNT162b2-Omi4/5 booster. In our study, the BCVax immunized with 3 injections (booster) also showed the pseudovirus neutralization activity against BA.2 and BA.4/5 in the comparable range ($10^3$ to $10^4$). These results suggest that BCVax has the potential as a booster and may provide similar pseudovirus neutralization potency against BA.5 to the bivalent booster vaccine BNT162b2-Omi4/5 (Fig 5D). Furthermore, as described earlier, vaccine with protein subunit as the antigen may offer safety advantage over other vaccine technologies.

Many publications have evaluated the safety and immunogenicity elicited by homologous (same type of vaccine) or heterologous (different type of vaccines) COVID-19 vaccine booster. The heterologous prime-boost were demonstrated to be well tolerated, and the neutralizing antibody induction can be further enhanced compared to homologous prime-boost subjects [53–55]. For example, Atmar et al. demonstrated that heterologous booster was able to induce 7–63 folds increase of neutralizing antibody, whereas homologous booster only showed 4–20 folds of increase in the study with different primary vaccination and booster combination [56]. In another study, Jin et al. demonstrated the Convidecia primed and ZF2001 boosted heterologous immunizations were safe with enhanced immunogenicity, supporting the utility of adenovirus-based vaccine with protein subunit based vaccine [54]. These findings that heterologous prime-boost strategy could provide better protectivity support BCVax as a new generation protein-based vaccine for enhancing protectivity against COVID-19 infection.

## Supporting information

**S1 Fig. Serum IgG titers of BCVax immunized BALB/c mice.** The delta S protein (10 μg) combined with or without adjuvant candidates of aluminum hydroxide plus CpG 1018 (50 μg plus 10 μg, triangle), AB801 (5 or 10 μg, square), or AB801-ISCOM (5 or 10 μg, circle, grey

bars) were immunized to BALB/c mice on day 0 and day 14. The anti-S protein IgG titer of (A) alpha, (B) beta, and (C) gamma strains in immunized mouse serum collected on day 28. Data were analyzed using one-way ANOVA.
(PDF)

**S1 File. Whole SDS-PAGE image of S protein.**
(PDF)

**S2 File. Animal study data set.**
(PDF)

**S3 File. Method protocols.**
(PDF)

## Acknowledgments

We thank the National RNAi Core Facility at Academia Sinica in Taiwan for providing pseudovirus neutralization assay services.

## Author Contributions

**Conceptualization:** Ming-Chen Yang, Ming-Tain Lai.

**Data curation:** Hsiao-Han Lin, Ping-Tzu Chu, Hsin-Yi Tung, Yi-Chen Wu, Juo-Ling Sun.

**Formal analysis:** Chun-Chung Wang, Wei-Chien Tang, Nan-Hsuan Wang, Yin-Chieh Kuo.

**Funding acquisition:** Michael Chang.

**Methodology:** Kuan-Ming Chen, Chu-Ying Chen, Yin-Cheng Hsieh, Sheng-Yu Liu.

**Supervision:** Wan-Fen Li, Wei-Han Lee, Jiann-Shiun Lai.

**Writing – original draft:** Ming-Chen Yang.

**Writing – review & editing:** Ming-Tain Lai.

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
