## [Decision Letter · Decision Letter 0]

16 Jan 2023

PONE-D-22-33711Development of a spike protein subunit-based COVID-19 vaccine with broad protection against various SARS-CoV-2 variantsPLOS ONE

Dear Dr. Lai,

Thank you for submitting your manuscript to PLOS ONE. After careful consideration, we feel that it has merit but does not fully meet PLOS ONE’s publication criteria as it currently stands. Therefore, we invite you to submit a revised version of the manuscript that addresses the points raised during the review process. I could obtain the comments from only one reviewer. Thus, I have carefully read your manuscript, and decided the minor revision. 

We look forward to receiving your revised manuscript.

Kind regards,

Etsuro Ito

Academic Editor

PLOS ONE

Journal Requirements:

3. During your revisions, please confirm whether the title is sufficiently specific, and update it in the manuscript file and online submission information if needed. Specifically, we note that this investigation was carried out in animals only (and not in humans), and we ask you to ensure that this is clearly reflected in your article title.

4. Please ensure you have specified in the Methods section of your manuscript text the origin of the SARS-CoV-2 spike protein used in this study, as well as the variants used in this study.

7. We note that you have included the phrase “data not shown” in your manuscript. Unfortunately, this does not meet our data sharing requirements. PLOS does not permit references to inaccessible data. We require that authors provide all relevant data within the paper, Supporting Information files, or in an acceptable, public repository. Please add a citation to support this phrase or upload the data that corresponds with these findings to a stable repository (such as Figshare or Dryad) and provide and URLs, DOIs, or accession numbers that may be used to access these data. Or, if the data are not a core part of the research being presented in your study, we ask that you remove the phrase that refers to these data.

8. Please include your full ethics statement in the ‘Methods’ section of your manuscript file. In your statement, please include the full name of the IRB or ethics committee who approved or waived your study, as well as whether or not you obtained informed written or verbal consent. If consent was waived for your study, please include this information in your statement as well. 

Reviewers' comments:

Reviewer's Responses to Questions

**Comments to the Author**

1. Is the manuscript technically sound, and do the data support the conclusions?

Reviewer #1: Yes

2. Has the statistical analysis been performed appropriately and rigorously? 

Reviewer #1: Yes

3. Have the authors made all data underlying the findings in their manuscript fully available?

Reviewer #1: Yes

4. Is the manuscript presented in an intelligible fashion and written in standard English?

Reviewer #1: Yes

5. Review Comments to the Author

Reviewer #1: The manuscript describes the design, biochemical characterization, and in vivo evaluation in mice of a novel vaccine targeting SARS-Cov-2. The vaccine consists of a protein antigen based on the S protein of the delta variant combined with an adjuvant (AB801-ISCOM). Two doses administered to mice resulted in relatively high titers of neutralizing antibodies when tested against a delta strain pseudovirus, with some neutralization against heterologous variants including omicron subvariants. Additionally a Tcell based immune response was also detected.

The manuscript meets the criterion of importance for publication given the continuing pandemic and an ongoing need for improved vaccines that show increased durability of immunity, and coverage of emerging viral variants. The data demonstrate that further investigation of BCVax either in additional preclinical studies, such as primate immunogenicity studies, or in the clinical setting is warranted to understand the translation of these studies in mice.

Here are some minor editorial suggestions –

- Some discussion of the rationale for choosing the delta sequence as the basis for BCVax is warranted, even if it is merely because that was the predominant variant in circulation when this work was initiated. Is the delta strain more likely to produce coverage of more recent variants than another based on its sequence similarity?

- Was the protein sequence of the antigen as shown in Figure 1 being compared to delta B.1.617.2? That is not clear from the Methods section or the Figure legend. Please add that information.

- The methods describe an ELISA for omicron BA.5 but results are only shown for the experiment in which a third booster was administered. Were samples from the 1st and 2nd administration not tested for anti-BA.5 neutralization? The neutralization response for BA.1 and BA.2 (particularly) were detectable but were modest and variable.

- In the methods I did not see a reference for production or purchase of the ACE2 protein.

- In the Discussion (Page 31) rather than use “protectivity” which implies protection against a virus challenge, I suggest using “neutralization in vitro” which is what was measured.

- The main drawback to mRNA based vaccines for SARS-Cov-2 is a lack of durable immune response. One potential advantage to this vaccine would be the Th1 response that is induced which may well result in greater durability of protection. This possibility is mentioned in the discussion briefly but should be expanded if possible. Are there data in the literature regarding Tcell based immunity resulting from mRNA-based vaccines (or lack thereof) to put these results in context? Reference 32 maybe?

- Although the manuscript is understandable as written, it would be worthwhile to have someone edit it for English word usage and grammar. For examples, in the Abstract it says “the protectivity of the available vaccines are often compromised” but should read “is compromised”, and the legend to Figure 2 says “receptivity” but should be “respectively”.

6. PLOS authors have the option to publish the peer review history of their article (what does this mean?). If published, this will include your full peer review and any attached files.

Reviewer #1: No

---

## [Author Response · Author response to Decision Letter 0]

1 Mar 2023

PONE-D-22-33711

Development of a spike protein subunit-based COVID-19 vaccine with broad protection against various SARS-CoV-2 variants

PLOS ONE

Editor’s comments:

Response: The manuscript has been revised to meet PLOS ONE’s style requirements, including title page format, fonts, line numbers, length of figure title, figure legend format, reference format, etc.

Response: Added the following descriptions to the Materials and Methods section (Animal immunization section, page 7, line 146-152):

Animal studies were conducted by Level Biotechnology Inc. CRO-Preclinical Testing Center, Taiwan. The procedures with animals described in this protocol have been reviewed by the test facility’s Institutional Animal Care and Use Committee (IACUC). The IACUC approval number was 210306-03 and 220511. During the study period, there is no need to give further anesthesia. At the end of study, all mice were sacrificed by CO2 euthanasia. All procedures described in protocol that involve study animals were conducted in a manner to avoid or minimize discomfort, distress, or pain to the animals, such as providing dietary supplement or 1 mg/kg Meloxicam if necessary. 

3. During your revisions, please confirm whether the title is sufficiently specific, and update it in the manuscript file and online submission information if needed. Specifically, we note that this investigation was carried out in animals only (and not in humans), and we ask you to ensure that this is clearly reflected in your article title.

Response: Thanks for the suggestion. The title as revised as follows:

The original title “Development of a spike protein subunit-based COVID-19 vaccine with broad protection against various SARS-CoV-2 variants” is changed to “Immunogenicity of a spike protein subunit-based COVID-19 vaccine with broad protection against various SARS-CoV-2 variants in animal studies”.

4. Please ensure you have specified in the Methods section of your manuscript text the origin of the SARS-CoV-2 spike protein used in this study, as well as the variants used in this study.

Response: The following descriptions in the Materials and Methods under Delta strain S protein expression and purification section is added in page 5, line 84-87):

The S protein DS2P employed in this manuscript (sequence ID: GenBank/ QWK65230.1, SARS-CoV-2 lineage B.1.617.2) were mutated at specific sites to abolish cellular furin cleavage and for protein stabilization in culture system to ensure the steady production. The mutation sites include T19R, G142D, ∆156-157, R158G, L452R, T478K, D614G, P681R, D950N.

5. In your Data Availability statement, you have not specified where the minimal data set underlying the results described in your manuscript can be found.

Response: The minimum data set is added as Supporting information S2 File.

6. PLOS requires an ORCID iD for the corresponding author in Editorial Manager on papers submitted after December 6th, 2016.

Response: The ORCID ID is 0000-0002-9829-7943.

7. We note that you have included the phrase “data not shown” in your manuscript. Unfortunately, this does not meet our data sharing requirements. PLOS does not permit references to inaccessible data. We require that authors provide all relevant data within the paper, Supporting Information files, or in an acceptable, public repository.

Response: The figure containing the data is included in Supporting information S1 Fig., as shown below.

8. Please include your full ethics statement in the ‘Methods’ section of your manuscript file. In your statement, please include the full name of the IRB or ethics committee who approved or waived your study, as well as whether or not you obtained informed written or verbal consent. If consent was waived for your study, please include this information in your statement as well.

Response: The following descriptions is added to the Materials and Methods under section of Animal immunization section (page 7, line 146-152): 

Animal studies were conducted by Level Biotechnology Inc. CRO-Preclinical Testing Center, Taiwan. The procedures with animals described in this protocol have been reviewed by the test facility's Institutional Animal Care and Use Committee (IACUC). The IACUC approval number was 210306-03 and 220511. During the study period, there is no need to give further anesthesia. At the end of study, all mice were sacrificed by CO2 euthanasia. All procedures described in protocol that involve study animals were conducted in a manner to avoid or minimize discomfort, distress, or pain to the animals, such as providing dietary supplement or 1 mg/kg Meloxicam if necessary. 

9. Please review your reference list to ensure that it is complete and correct. If you have cited papers that have been retracted, please include the rationale for doing so in the manuscript text, or remove these references and replace them with relevant current references.

Response: The reference list has been thoroughly reviewed and it is up to date, no retracted paper is found. During the review, a few more citations are added to the reference list to enhance relevant information.

The added references are:

Ref #19. Papa G, Mallery DL, Albecka A, Welch LG, Cattin-Ortola J, Luptak J, et al. Furin cleavage of SARS-CoV-2 Spike promotes but is not essential for infection and cell-cell fusion. PLoS Pathog. 2021;17(1):e1009246. Epub 2021/01/26. doi: 10.1371/journal.ppat.1009246. 

Ref #32. Ong SWX, Chiew CJ, Ang LW, Mak TM, Cui L, Toh M, et al. Clinical and virological features of SARS-CoV-2 variants of concern: a retrospective cohort study comparing B.1.1.7 (Alpha), B.1.315 (Beta), and B.1.617.2 (Delta). Clin Infect Dis. 2021. Epub 2021/08/24. doi: 10.1093/cid/ciab721.

Ref #33. Fisman DN, Tuite AR. Evaluation of the relative virulence of novel SARS-CoV-2 variants: a retrospective cohort study in Ontario, Canada. CMAJ. 2021;193(42):E1619-E25. Epub 2021/10/07. doi: 10.1503/cmaj.211248.

Ref #34. Twohig KA, Nyberg T, Zaidi A, Thelwall S, Sinnathamby MA, Aliabadi S, et al. Hospital admission and emergency care attendance risk for SARS-CoV-2 delta (B.1.617.2) compared with alpha (B.1.1.7) variants of concern: a cohort study. Lancet Infect Dis. 2022;22(1):35-42. Epub 2021/08/31. doi: 10.1016/S1473-3099(21)00475-8.

Ref #35. Wang L, Gao GF. The "Wolf" Is Indeed Coming: Recombinant "Deltacron" SARS-CoV-2 Detected. China CDC Wkly. 2022;4(14):285-7. Epub 2022/04/19. doi: 10.46234/ccdcw2022.054.

Ref #41. Moss P. The T cell immune response against SARS-CoV-2. Nat Immunol. 2022;23(2):186-93. Epub 2022/02/03. doi: 10.1038/s41590-021-01122-w.

Ref #48. Zhang Z, Mateus J, Coelho CH, Dan JM, Moderbacher CR, Galvez RI, et al. Humoral and cellular immune memory to four COVID-19 vaccines. Cell. 2022;185(14):2434-51 e17. Epub 2022/06/29. doi: 10.1016/j.cell.2022.05.022.

Ref #49. Hielscher F, Schmidt T, Klemis V, Wilhelm A, Marx S, Abu-Omar A, et al. NVX-CoV2373-induced cellular and humoral immunity towards parental SARS-CoV-2 and VOCs compared to BNT162b2 and mRNA-1273-regimens. J Clin Virol. 2022;157:105321.

Reviewer’s comments:

1. Some discussion of the rationale for choosing the delta sequence as the basis for BCVax is warranted, even if it is merely because that was the predominant variant in circulation when this work was initiated. Is the delta strain more likely to produce coverage of more recent variants than another based on its sequence similarity?

Response: The description on the rationale for selecting delta strain is expanded in Discussion section (page 26, line 465-469). The added description is as follows:

With the emergence of omicron strain in late 2021, it quickly became the dominant strain worldwide. Despite the fast spread, most of the omicron cases were found to have mild symptoms, while the delta strain is still the strain poses the highest risk of disease severity. Another potential threat of delta strain comes from the possibility of “deltacron” strain to emerge, especially an actual deltacron case was reported in China in Apr 2022.

With respect to potential broader coverage with BCVax, we did not have the result to support whether the immunization of delta S protein can provide better coverage than others against recent strains. There are 9 mutation sites within delta S protein, but there are 29 and 33 mutation sites within BA.2 and BA.5 S protein, respectively. There are only 3 mutations sites are the same between delta and BA.2 or BA.5 strain. 

2. Was the protein sequence of the antigen as shown in Figure 1 being compared to delta B.1.617.2? That is not clear from the Methods section or the Figure legend. Please add that information.

Response: As suggested, the following information is included in Material and Method under section of Delta strain S protein expression and purification (page 5, line 84-87): The S protein DS2P employed in this manuscript (sequence ID: GenBank/ QWK65230.1, SARS-CoV-2 lineage B.1.617.2) were mutated at specific sites to abolish cellular furin cleavage and for protein stabilization in culture system to ensure steady production. The mutation sites include T19R, G142D, ∆156-157, R158G, L452R, T478K, D614G, P681R, D950N. 

3. The methods describe an ELISA for omicron BA.5 but results are only shown for the experiment in which a third booster was administered. Were samples from the 1st and 2nd administration not tested for anti-BA.5 neutralization? The neutralization response for BA.1 and BA.2 (particularly) were detectable but were modest and variable.

Response: The analysis of IgG titer and pseudovirus neutralization activity against BA.5 after 1st and 2nd immunization was performed after we completed the initial evaluation. The results are included as shown below Fig.3B-3D. Although the IgG titers for BA.2 and BA.5 are comparable, the neutralization activity against BA.5 is lower than BA.2 which may be due to the additional mutation sites of Δ69-70, L452R, and F486V in BA.5 strain. This information is included the result section of the manuscript (page 16, line 312-313, 324-329).

Fig. 3B Fig. 3C 

Fig 3D.

4. In the methods I did not see a reference for production or purchase of the ACE2 protein.

Response: The source of purchased ACE2 is added in the Materials and Methods (under section of Binding affinity of delta S protein to human ACE2 receptor (page 6, line 116) as follows:

”ACE2 protein with Fc tag, purchased from Acro Biosystems, was first captured on Anti-Human IgG Fc Capture (AHC) biosensor”

5. In the Discussion (Page 31) rather than use “protectivity” which implies protection against a virus challenge, I suggest using “neutralization in vitro” which is what was measured.

Response: Thanks for the suggestion, the wording is revised as follow: 

In Discussion (page 27, line 479-480):

Although BCVax development is based on delta strain S protein, we found that the protectivity in vitro neutralization was not limited to delta strain only, with two or three injections of BCVax, the protectivity in vitro neutralization is effective to omicron strains as well.

In Discussion (page 29, line 529):

These results suggest that BCVax has the potential as a booster and may provide similar protectivity pseudovirus neutralization potency against BA.5 to the bivalent booster vaccine BNT162b2-Omi4/5 (Fig. 5D).

6. The main drawback to mRNA based vaccines for SARS-Cov-2 is a lack of durable immune response. One potential advantage to this vaccine would be the Th1 response that is induced which may well result in greater durability of protection. This possibility is mentioned in the discussion briefly but should be expanded if possible. Are there data in the literature regarding Tcell based immunity resulting from mRNA-based vaccines (or lack thereof) to put these results in context? Reference 32 maybe?

Response: Thanks for the suggestion, the strong Th1 response is indeed one of the strengths of BCVax immunization, as indicated in Fig.3, comparing to other adjuvant such as alum plus CpG, BCVax showed significant advantage on T cell response induction. The description in Discussion section (page 26-27, line 489-491, 504-512) is expanded as suggested. 

The expanded description on page 26, line 489-491:

It is reported that T cell memory encompasses broad recognition of viral proteins, the breadth of recognition can limit the impact of individual viral mutations and is likely to protect subjects from severe diseases form viral variants.

Regarding the comparison of mRNA and protein-based COVID-19 vaccine, the original Reference 32 only demonstrates the antibody and T cell immunity can be induced after different types of COVID-19 vaccine injection (mRNA-1273, BNT162b2, Ad26.COV2.S, and NVX-CoV2373), and the induced T cells are capable of recognizing different variants, but did not show the relative induction level of T cell induction among different vaccine types. We identified other references showing the comparison of different types of vaccines (shown in the figure and table below). Zhang et al. has published a study in 2022 to compare the humoral and cellular immune memory among four types of COVID-19 vaccines in human. The results showed that, at 6 months post immunization, antibody levels declined, while memory T cells and B cells were comparatively stable, and similar levels of CD4 and CD8 T cell responses were observed between mRNA and protein-based COVID-19 vaccines. Hielscher et al. published their study demonstrating NVX-CoV2373 vaccination profoundly induces spike-specific antibodies and CD4 T-cells responses, albeit at lower levels compared with BNT162b2 and mRNA-1273 vaccines, the cross reactivity of CD4 T-cells towards the parental strain and all VOCs tested still displayed the promising protectivity of the protein-based vaccine. These references are included into the manuscript as ref number 48 and 49, and the descriptions are added to Discussion section, page 27, line 504-512. 

 [Ref 48. Cell. 2022. PMID: 35764089] [Ref 49. J Clin Virol. 2022. PMID: 36279695]

7. Although the manuscript is understandable as written, it would be worthwhile to have someone edit it for English word usage and grammar. For examples, in the Abstract it says “the protectivity of the available vaccines are often compromised” but should read “is compromised”, and the legend to Figure 2 says “receptivity” but should be “respectively”.

Response: Thanks for the correction, revised accordingly. In addition, after further thoroughly reviewing the manuscript, we made a few additional changes as follows:

Page 3, Line 54: “protein subunit vaccine is known for its advantage on safety and stability” is changed to “protein subunit vaccines are known for their advantage on safety and stability.”

Page 8, Line 176-177: “and then stop solution (1N HCl).” is changed to “and then stop the reaction with 1N HCl.”

Page 23, Line 431: “CD8+GranzymeB+ populations were also enhanced after booster received” is changed to “CD8+GranzymeB+ populations were also enhanced after receiving the booster.”

---

## [Editor Report · Decision Letter 1]

9 Mar 2023

Immunogenicity of a spike protein subunit-based COVID-19 vaccine with broad protection against various SARS-CoV-2 variants in animal studies

PONE-D-22-33711R1

Dear Dr. Lai,

We’re pleased to inform you that your manuscript has been judged scientifically suitable for publication and will be formally accepted for publication once it meets all outstanding technical requirements.

Kind regards,

Etsuro Ito

Academic Editor

PLOS ONE

---

## [Editor Report · Acceptance letter]

16 Mar 2023

PONE-D-22-33711R1 

Immunogenicity of a spike protein subunit-based COVID-19 vaccine with broad protection against various SARS-CoV-2 variants in animal studies 

Dear Dr. Lai:

I'm pleased to inform you that your manuscript has been deemed suitable for publication in PLOS ONE. Congratulations! Your manuscript is now with our production department. 

Kind regards, 

on behalf of

Prof. Etsuro Ito 

Academic Editor

PLOS ONE